# Synthesis and Antiproliferative Evaluation of Novel Longifolene-Derived Tetralone Derivatives Bearing 1,2,4-Triazole Moiety

**DOI:** 10.3390/molecules25040986

**Published:** 2020-02-22

**Authors:** Xia-Ping Zhu, Gui-Shan Lin, Wen-Gui Duan, Qing-Min Li, Fang-Yao Li, Shun-Zhong Lu

**Affiliations:** 1School of Chemistry and Chemical Engineering, Guangxi University, Nanning 530004, China; xiapingzhu@163.com (X.-P.Z.); lqmgxu@163.com (Q.-M.L.); 2College of Pharmacy, Guilin Medical University, Guilin 541100, China; lifangyao@glmc.edu.cn; 3Guangxi Academy of Forestry, Nanning 530002, China; dwg105@gxu.edu.cn

**Keywords:** longifolene-derived tetralone, 1,2,4-triazole, antiproliferative activity, synthesis

## Abstract

Seventeen novel 2-(5-amino-1-(substituted sulfonyl)-1*H*-1,2,4-triazol-3-ylthio)-6- isopropyl-4,4-dimethyl-3,4-dihydronaphthalen-1(2*H*)-one compounds were synthesized from the abundant and naturally renewable longifolene and their structures were confirmed by FT-IR, NMR, and ESI-MS. The in vitro cytotoxicity of the synthesized compounds was evaluated by standard MTT assay against five human cancer cell lines, i.e., T-24, MCF-7, HepG2, A549, and HT-29. As a result, compounds **6d**, **6g**, and **6h** exhibited better and more broad-spectrum anticancer activity against almost all the tested cancer cell lines than that of the positive control, 5-FU. Some intriguing structure–activity relationships were found and are discussed herein by theoretical calculation.

## 1. Introduction

Tetralin (tetrahydronaphthalene) is a valuable ring for the construction of drugs. This ring structure is found in many naturally occurring plant-derived podophyllotoxin. The chemotherapeutic medications etoposide and teniposide are semisynthetic derivatives of podophyllotoxin, and are used as topoisomerase II inhibitor anticancer drugs [1]. Due to their high toxicity, drug resistance, and the gastrointestinal discomfort associated with their use, lots of structural modifications have been carried out; consequently, some derivatives with better performance, such as NK611, NPF, GL-331, and TOP53, have been discovered and tested in clinical trials [2]. The tetralin ring also exists in the structures of some clinically used anticancer drugs, for example, the anthracycline antibiotics doxorubicin, daunorubicin, epirubicin, and idarubicin [3], and so on. Many studies have been performed on tetralin derivatives for use as efficient antitumor agents [4,5,6,7,8,9].

Due to their diverse pharmacological properties, 1,2,4-triazole derivatives have been extensively studied for application in the fields of medicine and agrochemicals. Besides antimicrobial [10,11], anti-inflammatory [10,12], antituberculosis [13], antifungal [14], and herbicidal properties [15], antitumor activity has been reported [16], e.g., in the anticancer drugs letrozole, anastrozole, and vorozole [17]. Meanwhile, more and more new 1,2,4-triazole anticancer compounds are being studied [18,19]. In addition, many sulfonamide compounds with antitumor activity have been reported in recent years [20,21,22].

A tetralin derivative 7-isopropyl-1,1-dimethyl-1,2,3,4-tetrahydronaphthalene can be readily prepared from longifolene, which is a major constituent of the abundant and naturally renewable heavy turpentine oil, a by-product in the production of rosin and turpentine from living *Pinus*. In view of the factors mentioned above, and in continuation of our interest in seeking bioactive compounds from the biological source of natural products [23,24,25,26], a series of novel 2-(5-amino-1- (substituted sulfonyl)-1*H*-1,2,4-triazol-3-ylthio)-6-isopropyl-4,4-dimethyl-3,4-dihydronaphthalen-1 (2*H*)-one compounds were designed and synthesized. Additionally, structural characterization and antiproliferative evaluation in vitro against T-24, MCF-7, HepG2, A549, and HT-29 human cancer cell lines for all the target compounds were carried out.

## 2. Results and Discussion

### 2.1. Synthesis and Characterization

The synthetic route of compounds **6a–6q** is illustrated in Scheme 1. First, 7-Isopropyl-1,1-dimethyl-1,2,3,4-tetrahydronaphthalene (**2**) was prepared by isomerization and aromatization of longifolene, and further oxidized and brominated to give 4,4-dimethyl-2-bromo-6-isopropyl-3,4-dihydronaphthalen-1-one (**4**). Then, compound **4** was reacted with 5-amino-3-mercapto-1,2,4-triazole to obtain compound 2-(5-amino-1*H*-1,2,4-triazol-3ylthio)-6-isopropyl-4,4-dimethyl-3,4-dihydronaphthalen-1(*2H*)-one (**5**). Finally, 17 target compounds, i.e., **6a–6q**, were synthesized by the *N*-sulfonylation reaction of compound **5** with a series of substituted sulfonyl chlorides.

According to [27], compound **5** can exist in three tautomeric forms, namely, 1*H-*, 2*H-*, and 4*H*-forms; the 1*H*-form was preferred. The *N*-sulfonylation reaction process was monitored by HPLC, and only one new peak was observed, leading us to speculate that compound **5** is in the 1*H*-form. Then, a *N*-sulfonylation reaction may occur on the nitrogen atoms of the 1*H*-site and the amino group. Comparing the ^1^H-NMR spectra of compound **5** with the those of the target compounds, it was found that the signals of the amino groups were preserved, meaning that a *N*-sulfonylation reaction took place on the nitrogen atom of the 1*H*-site. The result was further confirmed by a nuclear overhauser effect spectroscopy (NOESY) experiment of the target compound **6a**. As shown in Figure 1, the hydrogen atoms of the amino group showed a stronger correlation with the hydrogen atoms of the methyl group on the phenylsulfonyl moiety than that of the methyl groups on the tetralone moiety, indicating that the amino group was closer to the phenylsulfonyl group than the methyl groups on the tetralone moiety.

The structures of the target compounds were confirmed by means of FT-IR, ^1^H-NMR, ^13^C-NMR, and ESI-MS. In the FT-IR spectra, the target compounds exhibited absorption bands at 3100–3465 cm^−1^, which were assigned to the N–H stretching vibration. The two strong absorption bands at 3000–2850 cm^−1^ and 1675 cm^−1^ were attributed to the stretching vibrations of the saturated C–H and C=O in the tetralone moiety, respectively. The absorption bands at about 1640 cm^−1^ were due to the vibrations of C=N in the 1,2,4-triazole moiety. The two absorption bands around 1160–1190 cm^−1^ and 1300–1360 cm^−1^, respectively, confirmed the presence of a sulfonyl group. In the ^1^H-NMR spectra of **6a**–**6q**, the signals shown at δ 6.0–8.5 ppm were assigned to the aromatic protons in the benzene rings and amino protons in the 1,2,4-triazole moiety. The other protons bonded to the saturated carbons of the tetralone moiety displayed signals in the range of δ 1.2–5.0 ppm. The ^13^C-NMR spectra of all the target compounds showed peaks for the carbons of C=O in tetralone moiety at about δ 193.2 ppm, and the other saturated carbons displayed signals in the region of δ 22–50 ppm. For the 1,2,4-triazole moiety, the signals at about δ 170.8 ppm and δ 151.7 ppm were assigned to the carbons of C=N. In addition, the molecular weights of the compounds **2**–**5** and target compounds **6a**–**6q** were confirmed by the ESI-MS (see Appendix A).

### 2.2. In Vitro Assay of Antiproliferative Activity

The in vitro antiproliferative activity of compounds **2**, **3**, **4**, **5**, and all 17 target compounds, i.e., **6a**–**6q**, against five human cancer cell lines (T-24 human bladder cancer cell, MCF-7 human breast cancer cell, HepG2 human liver cancer cell, A549 human lung adenocarcinoma drug-resistant cell line, and HT-29 human colon cancer cell line) were evaluated by MTT (methylthiazolytetrazolium) assay. In addition, the potent anticancer drug 5-fluorouracil (5-FU) was used as a positive control in each panel. The in vitro antiproliferative activity test results of compounds 2, 3, 4, 5, and all 17 target compounds **6a**–**6q** are shown in Table 2.

The target compounds presented obviously different antineoplastic activity against the five tested cancer cell lines. As shown in Table 2, compounds **6g**, **6h**, and **6d** exhibited excellent and broad-spectrum antitumor activity against almost all the tested cancer cell lines, and compounds **6b**, **6c**, and **6f** showed good activity against A549 and HT-29. All these compounds displayed better or comparable antitumor activity than that of the positive control, 5-FU. Notably, compound **6g** had the best antineoplastic activity against MCF-7, with IC_50_ values of 4.42 ± 2.93 µM, and compound **6h** against A549 with IC_50_ values of 9.89 ± 1.77 µM.

Some intriguing bioactive differences can be found. For example, compound **6h** (R = 3′-NO_2_-4′-Cl) exhibited obviously better antitumor activity than compound **6k** (R = 3′-NO_2_) and **6j** (R = 4′-Cl). Additionally, compound **6d** (R = 3′-Br) showed better antitumor activity than compound **6o** (R = 4′-Br). Compound **6c** (R = 2′-NO_2_) was better than compounds **6k** (R = 3′-NO_2_) and **6l** (R = 4′-NO_2_). Furthermore, compound **6g** (R = 2′, 3′, 4′, 5′, 6′-F) displayed better antineoplastic activity than compound **6p** (R = 2′, 4′-F). All these indicated that the position, type, and number of substituted groups significantly affects antitumor activity.

### 2.3. Theoretical Calculation and Analysis

The obvious contrast on antitumor activity between disubstituted compound **6h** (R = 3′-NO_2_-4′-Cl) and its corresponding monosubstituted compounds **6k** (R = 3′-NO_2_) and **6j** (R = 4′-Cl) attracted our attention. According to the frontier molecular orbital theory, HOMO has the ability to provide electrons, while LUMO readily accepts electrons; these two frontier orbitals affect the bioactivity of compounds [28]. Therefore, the frontier molecular orbitals of compounds **6h** (R = 3′-NO_2_-4′-Cl), **6k** (R = 3′-NO_2_), and **6j** (R = 4′-Cl) were calculated by means of DFT/B3LYP/6-31G (d, p) [28] in the Gaussian 09 package [29] on the Computer Supercomputing Platform at Guangxi University, and the result was viewed using the GaussView 5 software [30]. 

As shown in Figure 2a,d,g, a large portion of the HOMO was located on the 1,2,4-triazole ring, the amino group, the sulfur atom of thioether group, and the oxygen atom of carbonyl group for three compounds. However, the LUMO for compounds **6h** (R = 3′-NO_2_-4′-Cl) and **6k** (R = 3′-NO_2_) was located on the substituted phenylsulfonyl group moieties (Figure 2b,e), and showed phase reversal. The compound **6j** (R = 4′-Cl) had LUMO on the tetralone moiety, showing an obvious difference with compounds **6h** and **6k**. In addition, molecule charge distribution was also one of the influential factors concerning activity. So, the electrostatic potentials (ESPs) and dipole moments for three compounds were calculated by means described above (Figure 2c,f,i). In comparison to compounds **6k** and **6j**, **6h** showed higher negative electrostatic potential on the end of the R group, and possessed higher dipole moment, i.e., 9.48 D, than compounds **6k**, i.e., 9.01 D, and **6j**, 5.23 D. Overall, these differences were potentially responsible for the contrast of compounds **6h**, **6k**, and **6j** in terms of antitumor activity. It would be useful to further investigate these compounds.

## 3. Experimental Section

### 3.1. General Information

The structures of the synthesized derivatives of tetralin were confirmed by means of ^1^H-NMR, ^13^C-NMR, FT-IR, and ESI-MS. NMR spectra were recorded in a CDCl_3_ or DMSO-*d_6_* solvent on a Bruker Avance III HD 600 MHz spectrometer (Bruker Co., Ltd., Zurich, Switzerland). FT-IR spectra were recorded as KBr pellets on Nicolet iS50 FT-IR spectrometer (Thermo Scientific Co., Ltd., Madison, WI., USA). MS spectra were obtained by means of the electrospray ionization (ESI) method on TSQ Quantum Access MAX HPLC-MS instrument (Thermo Scientific Co., Ltd., Waltham, MA, USA). Melting points were determined using a MP420 automatic melting point apparatus (Hanon Instruments Co., Ltd., Jinan, China) and were uncorrected. The GC analysis was conducted on an Agilent 6890 GC (Agilent Technologies Inc., Santa Clara, CA., USA) equipped with column HP-1 (30 m, 0.530 mm, 0.88 µm) and FID. The HPLC analysis was performed on a Waters 1525 instrument (Waters Co., Ltd., USA) equipped with column SunFire C18 5 µm (4.6 mm × 150 mm). Longifolene (GC purity 65%) was provided by Wuzhou Pine Chemicals Co., Ltd. Wuzhou, Guangxi, China. Other reagents were provided by commercial suppliers, and were used as received. 

### 3.2. Preparation of Catalyst

A catalyst was prepared according to a method described in the literature [31]. After diluting with n-propanol (30% by weight), zirconium was used as the precursor. Sulfuric acid (1.02 mL) was added to the zirconium precursor, and water was then added dropwise under continuous stirring until the formation of gel occurred (4.2 mL, water to propoxide molar ratio = 2.7). The resulting gel was dried at 110 °C for 12 h, followed by calcination at 600 °C for 2 h in a static air atmosphere to yield the sulfated zirconia catalyst.

### 3.3. Synthesis of 7-isopropyl-1,1-dimethyl-1,2,3,4-tetrahydronaphthalene *(**2**)*

Compound **2** was prepared according to a method described in the literature [32]. Longifolene (10 g, 49 mmol) and the catalyst (1 g) were mixed and stirred for 1h at 180 °C. The reaction mixture was then filtered to separate the catalyst. The filtrate was separated by column chromatography using silica gel and petroleum ether to obtain a colorless liquid, compound **2**, at a 54.6% yield. The characterization data of compound **2** was consistent with that reported in the literature.

### 3.4. Synthesis of 6-isopropyl-4,4-dimethyl-3,4-dihydronaphthalen-1(2H)-one *(**3**)*

Compound **2** (10 g, 46 mmol), CuCl_2_ (0.62 g, 4.6 mmol), and CH_3_CN (20 mL) were mixed and stirred for 30 min at room temperature. Then, the mixed solution of 25 mL TBHP (70% in water, wt%) (185 mmol) and CH_3_CN (25 mL) was added to the reaction system. Afterwards, the reaction mixture was heated to 42 °C and stirred for 24 h. The reaction process was monitored by TLC. When the reaction was completed, the reaction mixture was cooled to room temperature. The unreacted TBHP was destroyed with an appropriate amount of Na_2_S_2_O_3_. Then, the reaction mixture was extracted three times with ethyl acetate. The combined organic phase was washed with saturated sodium chloride and deionized water, respectively. Finally, the organic layer was concentrated and further purified by silica gel column chromatography using EtOAc/petroleum ether (1:60, *v*/*v*) as an eluent to obtain a colorless liquid, compound **3**. Yield: 69.2%. The characterization data of compound **3** was consistent with that reported in the literature [33].

### 3.5. Synthesis of 2-bromo-6-isopropyl-4,4-dimethyl-3,4-dihydronaphthalen-1(2H)-one *(**4**)*

Compound **3** (2.96 g, 10 mmol), NBS (1.87 g, 10.50 mmol), and NH_4_OAc (8 mg, 1 mmol) was added into dry Et_2_O (15 mL), respectively [34]. After stirring at room temperature for 30 min, the mixture was filtered. The filtrate was washed with water, dried, and evaporated to remove the Et_2_O. The residue was purified by silica gel column chromatography using EtOAc/petroleum ether (1:40, *v*/*v*) as an eluent to give a white solid, i.e., compound **4**. Yield: 88.5%; m.p. 112.3–113.9 °C; FT-IR (KBr, cm^−1^): 3067 (Ar–H), 1694 (C=O), 1602, 1459 (Ar); ^1^H-NMR (600 MHz, CDCl_3_) δ: 8.03 (d, *J* = 8.1 Hz, 1H, C_8_–H), 7.25–7.21 (m, 2H, C_7_–H, C_5_–H), 5.10 (dd, *J* = 13.5, 5.4 Hz, 1H, C_2_–H), 2.97 (m, 1H, C_11_–H), 2.64–2.51 (m, 2H, C_3_–H), 1.50 (s, 3H, C_4_–CH_3_), 1.43 (s, 3H, C_4_–CH_3_), 1.29 (d, *J* = 6.9 Hz, 6H, C_11_–CH_3_); ^13^C-NMR (151 MHz, CDCl_3_) δ: 190.63, 156.17, 151.46, 128.89, 127.55, 125.28, 123.90, 50.61, 48.97, 36.89, 34.65, 30.35, 29.92, 23.63; MS (ESI) *m*/*z* = 297.01 ([M + H^+^]).

### 3.6. Synthesis of 2-(5-amino-1H-1,2,4-triazol-3-ylthio)-6-isopropyl-4,4-dimethyl-3,4-dihydronaphthalen-1 (2H)-one *(**5**)*

First, 5-Amino-3-mercapto-1,2,4-triazole (0.50 g, 4.30 mmol) was dissolved in a mixture of KOH (0.29 g, 0.66 mmol) in EtOH (20 mL) and stirred at room temperature for 30 min. Subsequently, the solution of compound **4** (1.18 g, 3.60 mmol) in EtOH (20 mL) was added slowly. Then, the reaction mixture was refluxed at 80 °C for 4 h under stirring. The reaction process was monitored by TLC. When the reaction was complete, the solvent was removed by rotary evaporation. Afterwards, the residual mixture was neutralized with dilute acetic acid and extracted three times with EtOAc. Finally, the combined organic phase was concentrated and purified by silica gel column chromatography using EtOAc/petroleum ether (1:1, *v*/*v*) as an eluent to afford a white solid, i.e., compound **5**. Yield: 87.3%; m.p. 195.2-195.7 °C; FT-IR (KBr, cm^−1^): 3417, 3336, 3292, 3179 (N–H), 1682 (C=O), 1636 (C=N), 1602, 1486 (Ar); ^1^H-NMR (600 MHz, DMSO-*d_6_*) δ: 11.94 (s, 1H,-NH), 7.79 (d, *J* = 8.1 Hz, 1H, C_8_–H), 7.39 (s, 1H, C_5_–H), 7.25 (d, *J* = 8.1 Hz, 1H, C_7_–H), 6.05 (s, 2H, -NH_2_), 4.85 (d, *J* = 9.8 Hz, 1H, C_2_–H), 3.02–2.90 (m, 1H, C_11_–H), 2.38–2.23 (m, 2H, C_3_–H), 1.38 (d, *J* = 9.1 Hz, 6H, C_4_–CH_3_), 1.22 (d, *J* = 6.9 Hz, 6H, C_11_–CH_3_); ^13^C-NMR (151 MHz, DMSO-*d_6_*) δ: 193.87, 157.73, 155.69, 154.14, 152,13, 128.85, 127.71, 125.07, 124.81, 49.75, 45.40, 35.72, 34.29, 30.62, 29.65, 23.96, 23.94; MS (ESI) *m*/*z* = 331.08 ([M + H^+^]).

### 3.7. General Procedure for the Synthesis of the Target Compounds *(**6a**–**6q**)*

Compound **5** (0.40 g, 1.20 mmol) and NaHCO_3_ (0.13 g, 1.50 mmol) in dry acetonitrile (10 mL) were mixed and stirred for 15 min at room temperature. Then, a solution of sulfonyl chloride (1.5 mmol) in acetonitrile (10 mL) was slowly added dropwise. Afterwards, the reaction mixture was stirred for 24 h at 40 °C. The reaction was quenched with water (20 mL), and the reaction mixture was extracted three times with CH_2_Cl_2_. The combined organic phase was washed three times with deionized water, dried over anhydrous Na_2_SO_4_, and distilled to remove the CH_2_Cl_2_. The crude product was purified by silica gel column chromatography using EtOAc/petroleum ether (1:5, *v*/*v*) as an eluent to afford the target products **6a**–**6q**. Yield: 85%–90%.

*2-((5-Amino-1-((2,4,6-trimethylphenyl)sulfonyl)-1H-1,2,4-triazol-3-yl)thio)-6-isopropyl-4,4-dimethyl-3,4-dihydronaphthalen-1(2H)-one* (**6a**): White solid, m.p. 193.8–194.2 °C; FT-IR (KBr, cm^−1^): 3463, 3280, 3201, 3102 (N–H), 1674 (C=O), 1639 (C=N), 1602, 1554, 1492 (Ar), 1358, 1170 (O=S=O), 606 (C–S–C); ^1^H-NMR (600 MHz, CDCl_3_) δ: 7.90 (d, *J* = 8.1 Hz, 1H, C_8_–H), 7.19 (s, 1H, C_5_–H), 7.18–7.15 (m, 1H, C_7_–H), 6.96 (s, 2H, C_20_–H, C_22_–H), 6.33 (d, *J* = 27.6 Hz, 2H, -NH_2_), 4.76 (dd, *J* = 13.6, 5.0 Hz, 1H, C_2_–H), 2.99–2.90 (m, 1H, C_11_–H), 2.55 (s, 6H, C_19_–CH_3_, C_23_–CH_3_), 2.32 (s, 3H, C_21_–CH_3_), 2.29-2.19 (m, 2H, C_3_–H), 1.34 (s, 3H, C_4_–CH_3_),), 1.27 (t, *J* = 8.3 Hz, 6H, C_11_–CH_3,_ C_4_–CH_3_); ^13^C-NMR (151 MHz, CDCl_3_) δ: 193.17, 160.25, 156.04, 155.81, 151.66, 145.02, 141.29, 132.30, 131.02, 128.51, 128.19, 124.86, 123.91, 50.12, 45.01, 35.48, 34.63, 30.35, 29.18, 23.68, 23.66, 22.76, 21.15; MS (ESI) *m*/*z* = 513.14 ([M + H^+^]).

*2-((5-Amino-1-((2,5-dimethylphenyl)sulfonyl)-1H-1,2,4-triazol-3-yl)thio)-6-isopropyl-4,4-dimethyl-3,4-dihydronaphthalen-1(2H)-one* (**6b**): White solid, m.p. 182.1–182.3 °C; FT-IR (KBr, cm^−1^): 3457, 3297, 3212, 3138 (N–H), 3028 (Ar–H), 1678 (C=O), 1649 (C=N), 1601, 1557, 1486 (Ar), 1364, 1180 (O=S=O), 620 (C–S–C); ^1^H-NMR (600 MHz, DMSO-*d_6_*) δ: 7.86 (s, 1H, C_23_–H), 7.74 (d, *J* = 8.1 Hz, 1H, C_8_–H), 7.50 (d, *J* = 7.3 Hz, 1H, C_20_–H), 7.45 (s, 2H, -NH_2_), 7.37 (d, *J* = 4.9 Hz, 2H, C_5_–H, C_21_–H), 7.24 (d, *J* = 8.1 Hz, 1H, C_7_–H), 4.86 (dd, *J* = 13.3, 5.3 Hz, 1H, C_2_–H), 2.95 (m, 1H, C_11_–H), 2.40 (s, 3H, C_22_–CH_3_), 2.35 (s, 3H, C_19_–CH_3_), 2.20-2.10 (m, 2H, C_3_–H), 1.27 (s, 3H, C_4_–CH_3_), 1.24 (s, 3H, C_4_–CH_3_), 1.21 (d, *J* = 6.9 Hz, 6H, C_11_–CH_3_); ^13^C-NMR (151 MHz, DMSO-*d_6_*) δ: 193.15, 160.87, 157.53, 155.89, 152.04, 137.12, 136.29, 135.46, 135.21, 133.48, 130.53, 128.54, 127.69, 125.09, 124.82, 49.96, 44.70, 35.78, 34.30, 30.44, 28.96, 23.91, 20.73, 19.38; MS (ESI) *m*/*z* = 501.18 ([M + H^+^]).

*2-((5-Amino-1-((2-nitrophenyl)sulfonyl)-1H-1,2,4-triazol-3-yl)thio)-6-isopropyl-4,4-dimethyl-3,4-dihydronaphthalen-1(2H)-one* (**6c**): White solid, m.p. 182.2–183.9 °C; FT-IR (KBr, cm^−1^): 3453, 3324, 3236, 3101 (N–H), 1681 (C=O), 1665 (C=N), 1602, 1571, 1481 (Ar), 1548,1263 (-NO_2_), 1372, 1190 (O=S=O), 615 (C–S–C); ^1^H-NMR (600 MHz, DMSO*-d_6_*) δ: 8.17 (dd, *J* = 8.0, 1.1 Hz, 1H, C_20_–H), 8.07 (m, 1H, C_22_–H), 7.98 (d, *J* = 7.9 Hz, 1H, C_23_–H), 7.93 (m, 1H, C_21_–H), 7.75 (d, *J* = 8.1 Hz, 1H, C_8_–H), 7.49 (s, 2H, -NH_2_), 7.38 (d, *J* = 1.2 Hz, 1H, C_5_–H), 7.26 (d, *J* = 8.2 Hz, 1H, C_7_–H), 4.89 (dd, *J* = 13.6, 5.1 Hz, 1H, C_2_–H), 2.97 (m, 1H, C_11_–H), 2.25–2.13 (m, 2H, C_3_–H), 1.29 (s, 3H, C_4_–CH_3_), 1.27 (s, 3H, C_4_–CH_3_), 1.22 (d, *J* = 6.9 Hz, 6H, C_11_–CH_3_); ^13^C-NMR (151 MHz, DMSO*-d_6_*) δ: 193.16, 162.06, 157.80, 155.89, 152.00, 147.68, 137.50, 133.88, 131.97, 129.31, 128.64, 127.72, 126.08, 125.12, 124.80, 49.96, 44.70, 35.82, 34.30, 30.51, 29.23, 23.93; MS (ESI) *m*/*z* = 516.17 ([M + H^+^]).

*2-((5-Amino-1-((3-bromophenyl)sulfonyl)-1H-1,2,4-triazol-3-yl)thio)-6-isopropyl-4,4-dimethyl-3,4-dihydronaphthalen-1(2H)-one* (**6d**): White solid, m.p. 185.6–186.0 °C; FT-IR (KBr, cm^−1^): 3457, 3293, 3212, 3100 (N–H), 1679 (C=O), 1649 (C=N), 1602, 1561, 1487 (Ar), 1374, 1185 (O=S=O), 617 (C–S–C); ^1^H-NMR (600 MHz, DMSO*-d_6_*) δ: 8.11 (s, 1H, C_19_–H), 8.05 (d, *J* = 7.2 Hz, 1H, C_23_–H), 7.90 (d, *J* = 7.9 Hz, 1H, C_21_–H), 7.78 (d, *J* = 8.1 Hz, 1H, C_8_–H), 7.67 (t, *J* = 8.0 Hz, 1H, C_22_–H), 7.59 (s, 2H, -NH_2_), 7.40 (s, 1H, C_5_–H), 7.26 (d, *J* = 7.9 Hz, 1H, C_7_–H), 4.94 (dd, *J* = 13.8, 4.8 Hz, 1H, C_2_–H), 3.02-2.90 (m, 1H, C_11_–H), 2.29-2.16 (m, 2H, C_3_–H), 1.41 (s, 3H, C_4_–CH_3_), 1.37 (s, 3H, C_4_–CH_3_), 1.22 (dd, *J* = 6.0, 5.1 Hz, 6H, C_11_–CH_3_); ^13^C-NMR (151 MHz, DMSO*-d_6_*) δ: 193.14, 162.38, 158.12, 155.93, 152.00, 138.68, 138.33, 132.68, 130.04, 128.64, 127.72, 127.01, 125.13, 124.90, 123.07, 49.97, 44.80, 35.91, 34.31, 30.71, 29.28, 23.94; MS (ESI) *m*/*z* = 551.07 ([M + H^+^]).

*2-((5-Amino-1-((4-(trifluoromethyl)phenyl)sulfonyl)-1H-1,2,4-triazol-3-yl)thio)-6-isopropyl-4,4-dimethyl-3,4-dihydronaphthalen-1(2H)-one* (**6e**): White solid, m.p. 185.5–189.7 °C; FT-IR (KBr, cm^−1^): 3472, 3306, 3216, 3101 (N–H), 1677 (C=O), 1654 (C=N), 1603, 1563, 1490 (Ar), 1385, 1192 (O=S=O), 1322 (–CF_3_), 624 (C–S–C); ^1^H-NMR (600 MHz, CDCl_3_) δ: 8.06 (d, *J* = 8.3 Hz, 2H, C_19_–H, C_23_–H), 7.93 (d, *J* = 8.1 Hz, 1H, C_8_–H), 7.81 (d, *J* = 8.4 Hz, 2H, C_20_–H, C_22_–H), 7.25 (d, *J* = 1.5 Hz, 1H, C_5_–H), 7.21 (dd, *J* = 8.1, 1.5 Hz, 1H, C_7_–H), 6.31 (s, 2H, -NH_2_), 4.82 (dd, *J* = 14.0, 4.7 Hz, 1H, C_2_–H), 2.97 (m, 1H, C_11_–H), 2.38–2.23 (m, 2H, C_3_–H), 1.46 (s, 3H, C_4_–CH_3_), 1.41 (s, 3H, C_4_–CH_3_), 1.28 (d, *J* = 6.9 Hz, 6H, C_11_–CH_3_); ^13^C-NMR (151 MHz, CDCl_3_) δ: 193.02, 162.34, 156.68, 156.05, 151.54, 140.04, 136.72, 136.50, 136.28, 136.08, 128.72, 128.54, 128.19, 126.66, 125.04, 124.07, 123.76, 121.95, 50.04, 44.83, 35.67, 34.65, 30.56, 29.39, 23.67, 23.65; MS (ESI) *m*/*z* = 539.15 ([M + H^+^]).

*2-((5-Amino-1-((4-methoxyphenyl)sulfonyl)-1H-1,2,4-triazol-3-yl)thio)-6-isopropyl-4,4-dimethyl-3,4-dihydronaphthalen-1(2H)-one* (**6f**): White solid, m.p. 184.8–185.5 °C; FT-IR (KBr, cm^−1^): 3485, 3302, 3232, 3171, 3171 (N–H), 1679 (C=O), 1628 (C=N), 1604, 1591, 1500, 1469 (Ar), 1363, 1155 (O=S=O), 605 (C–S–C); ^1^H-NMR (600 MHz, DMSO*-d_6_*) δ: 7.85 (dd, *J* = 6.9, 2.0 Hz, 2H, C_19_–H, C_23_–H), 7.78 (d, *J* = 8.0 Hz, 1H, C_8_–H), 7.43 (s, 2H, -NH_2_), 7.41 (s, 1H, C_5_–H), 7.27 (d, *J* = 8.1 Hz, 1H, C_7_–H), 7.18 (d, *J* = 9.0 Hz, 2H, C_20_–H, C_22_–H), 4.92 (dd, *J* = 13.8, 4.3 Hz, 1H, C_2_–H), 3.87 (s, 3H, -OCH_3_), 2.96 (m, 1H, C_11_–H), 2.20 (m, 2H, C_3_–H), 1.40 (s, 3H, C_4_–CH_3_), 1.36 (s, 3H, C_4_–CH_3_), 1.24-1.20 (m, 6H, C_11_–CH_3_); ^13^C-NMR (151 MHz, DMSO*-d_6_*) δ: 193.25, 164.82, 161.41, 157.90, 155.90, 152.06, 130.47, 128.69, 127.81, 127.73, 125.12, 124.84, 115.57, 56.47, 49.95, 44.80, 35.87, 34.32, 30.59, 29.31, 23.94; MS (ESI) *m*/*z* = 501.18 ([M + H^+^]). 

*2-((5-Amino-1-((perfluorophenyl)sulfonyl)-1H-1,2,4-triazol-3-yl)thio)-6-isopropyl-4,4-dimethyl-3,4-dihydronaphthalen-1(2H)-one* (**6g**): White solid, m.p. 189.3–190.6 °C; FT-IR (KBr, cm^−1^): 3455, 3304, 3226, 3154 (N–H), 1689 (C=O), 1651 (C=N), 1606, 1570, 1522, 1508 (Ar), 1278, 1200 (O=S=O), 618 (C–S–C); ^1^H-NMR (600 MHz, CDCl_3_) δ: 7.90 (d, *J* = 8.1 Hz, 1H, C_8_–H), 7.25 (d, *J* = 1.2 Hz, 1H, C_5_–H), 7.20 (dd, *J* = 8.1, 1.4 Hz, 1H, C_7_–H), 6.34 (s, 2H, -NH_2_), 4.87 (dd, *J* = 14.1, 4.6 Hz, 1H, C_2_–H), 3.04-2.91 (m, 1H, C_11_–H), 2.48-2.27 (m, 2H, C_3_–H), 1.45 (d, *J* = 8.3 Hz, 6H, C_4_–CH_3_), 1.29 (d, *J* = 6.9 Hz, 6H, C_11_–CH_3_); ^13^C-NMR (151 MHz, CDCl_3_) δ: 192.92, 163.53, 156.52, 156.08, 151.63, 146.34, 144.53, 144.45, 138.89, 137.18, 128.36, 128.00, 124.97, 124.13, 50.20, 44.86, 35.75, 34.64, 30.59, 29.07, 23.64, 23.61; MS (ESI) *m*/*z* = 561.03 ([M + H^+^]).

*2-((5-Amino-1-((4-chloro-3-nitrophenyl)sulfonyl)-1H-1,2,4-triazol-3-yl)thio)-6-isopropyl-4,4-dimethyl-3,4-dihydronaphthalen-1(2H)-one* (**6h**): White solid, m.p. 176.9–178.6 °C; FT-IR (KBr, cm-1): 3443, 3316, 3232, 3106 (N–H), 1684 (C=O), 1658 (C=N), 1604, 1573, 1500, 1489 (Ar), 1387 (O=S=O), 1533, 1342 (-NO_2_), 1189 (O=S=O), 614 (C–S–C); ^1^H-NMR (600 MHz, DMSO-*d_6_*) δ: 8.68 - 8.59 (m, 1H, C_19_–H), 8.19-8.13 (m, 1H, C_23_–H), 8.12 (t, *J* = 4.7 Hz, 1H, C_22_–H), 7.76 (d, *J* = 8.1 Hz, 1H, C_8_–H), 7.64 (s, 2H, -NH_2_), 7.41 (s, 1H, C_5_–H), 7.26 (d, *J* = 8.1 Hz, 1H, C_7_–H), 4.94 (dd, *J* = 13.9, 4.7 Hz, 1H, C_2_–H), 3.04–2.90 (m, 1H, C_11_–H), 2.34-2.14 (m, 2H, C_3_–H), 1.38 (d, *J* = 15.8 Hz, 6H, C_4_–CH_3_), 1.23 (dd, *J* = 6.9, 2.0 Hz, 6H, C_11_–CH_3_); ^13^C-NMR (151 MHz, DMSO-*d_6_*) δ: 193.10, 162.89, 158.07, 155.97, 152.04, 148.04, 136.22, 134.40, 133.03, 132.58, 128.61, 127.67, 125.54, 125.14, 124.95, 49.98, 44.82, 35.91, 34.31, 30.64, 29.24, 23.93; MS (ESI) *m*/*z* = 551.07 ([M + H^+^]).

*2-((5-Amino-1-(phenylsulfonyl)-1H-1,2,4-triazol-3-yl)thio)-6-isopropyl-4,4-dimethyl-3,4-dihydronaphthalen-1(2H)-one* (**6i**): White solid, m.p. 217.0–217.5 °C; FT-IR (KBr, cm^−1^): 3454, 3291, 3238.74, 3174 (N–H), 3065 (Ar–H), 1674 (C=O), 1627 (C=N), 1603, 1560, 1464 (Ar), 1375, 1187 (O=S=O), 614 (C–S–C); ^1^H-NMR (600 MHz, DMSO-*d_6_*) δ: 7.91 (d, *J* = 7.5 Hz, 2H, C_19_–H, C_23_–H), 7.84 (t, *J* = 7.5 Hz, 1H, C_21_–H), 7.78 (d, *J* = 8.1 Hz, 1H, C_8_–H), 7.70 (t, *J* = 7.9 Hz, 2H, C_20_–H, C_22_–H), 7.51 (s, 2H, -NH_2_), 7.41 (s, 1H, C_5_–H), 7.27 (d, *J* = 8.2 Hz, 1H, C_7_–H), 4.92 (dd, *J* = 13.7, 4.8 Hz, 1H, C_2_–H), 3.03-2.93 (m, 1H, C_11_–H), 2.25-2.11 (m, 2H, C_3_–H), 1.40 (s, 3H, C_4_–CH_3_), 1.35 (s, 3H, C_4_–CH_3_), 1.22 (t, *J* = 6.3 Hz, 6H, C_11_–CH_3_); ^13^C-NMR (151 MHz, DMSO-*d_6_*) δ: 193.23, 161.86, 158.11, 155.92, 152.05, 136.60, 135.80, 130.41, 128.70, 127.90, 127.74, 125.13, 124.86, 49.91, 44.71, 35.89, 34.32, 30.62, 29.31, 23.95; MS (ESI) *m*/*z* = 470.07 ([M + H^+^]).

*2-((5-Amino-1-((4-chlorophenyl)sulfonyl)-1H-1,2,4-triazol-3-yl)thio)-6-isopropyl-4,4-dimethyl-3,4-dihydronaphthalen-1(2H)-one* (**6j**): White solid, m.p. 205.5–207.1 °C; FT-IR (KBr, cm^−1^): 3468, 3300, 3216, 3097 (N–H), 1677 (C=O), 1652 (C=N), 1602, 1561, 1488 (Ar), 1383, 1182 (O=S=O), 632 (C–S–C); ^1^H-NMR (600 MHz, DMSO*-d_6_*) δ: 7.91 (d, *J* = 8.5 Hz, 2H, C_19_–H, C_23_–H), 7.82-7.73 (m, 3H, C_20_–H, C_22_–H, C_8_–H), 7.55 (s, 2H, -NH_2_), 7.41 (s, 1H, C_5_–H), 7.27 (d, *J* = 8.1 Hz, 1H, C_7_–H), 4.91 (dd, *J* = 13.3, 5.3 Hz, 1H, C_2_–H), 2.97 (m, 1H, C_11_–H), 2.19 (t, *J* = 9.8 Hz, 2H, C_3_–CH), 1.36 (d, *J* = 15.6 Hz, 6H, C_4_–CH_3_), 1.22 (d, *J* = 6.9 Hz, 6H, C_11_–CH_3_); ^13^C-NMR (151 MHz, DMSO*-d_6_*) δ: 193.14, 162.18, 158.11, 155.92, 152.03, 140.91, 135.30, 130.61, 129.86, 128.67, 127.72, 125.14, 124.87, 49.94, 44.73, 35.87, 34.31, 30.59, 29.29, 23.94; MS (ESI) *m*/*z* = 505.05 ([M + H^+^]).

*2-((5-Amino-1-((3-nitrophenyl)sulfonyl)-1H-1,2,4-triazol-3-yl)thio)-6-isopropyl-4,4-dimethyl-3,4-dihydronaphthalen-1(2H)-one* (**6k**): White solid, m.p. 196.4–197.9 °C; FT-IR (KBr, cm^−1^): 3454, 3313, 3226, 3103 (N–H), 1685 (C=O), 1658 (C=N), 1606, 1569, 1488 (Ar), 1533, 1351 (-NO_2_), 1378, 1189 (O=S=O), 615 (C–S–C); ^1^H-NMR (600 MHz, DMSO*-d_6_*) δ: 8.64 (d, *J* = 6.7 Hz, 2H, C_21_–H, C_23_–H), 8.32 (d, *J* = 6.0 Hz, 1H, C_19_–H), 8.01 (t, *J* = 8.4 Hz, 1H, C_22_–H), 7.75 (d, *J* = 8.1 Hz, 1H, C_8_–H), 7.67 (s, 2H, -NH_2_), 7.41 (s, 1H, C_5_–H), 7.25 (d, *J* = 7.4 Hz, 1H, C_7_–H), 4.95 (dd, *J* = 14.0, 4.4 Hz, 1H, C_2_–H), 2.97 (t, *J* = 9.9 Hz, 1H, C_11_–H), 2.22 (dd, *J* = 40.3, 26.8 Hz, 2H, C_3_–H), 1.43 (s, 3H, C_4_–CH_3_), 1.36 (s, 3H, C_4_–CH_3_), 1.22 (d, *J* = 6.8 Hz, 6H, C_11_–CH_3_); ^13^C-NMR (151 MHz, DMSO*-d_6_*) δ: 193.11, 162.82, 158.13, 155.96, 152.03, 148.55, 137.84, 133.76, 132.59, 130.28, 128.61, 127.66, 125.12, 124.93, 122.84, 49.99, 44.81, 35.92, 34.31, 30.64, 29.21, 23.93; MS (ESI) *m*/*z* = 516.16 ([M + H^+^]).

*2-((5-Amino-1-((4-nitrophenyl)sulfonyl)-1H-1,2,4-triazol-3-yl)thio)-6-isopropyl-4,4-dimethyl-3,4-dihydronaphthalen-1(2H)-one* (**6l**): White solid, m.p. 199.7–200.3 °C; FT-IR (KBr, cm^−1^): 3475, 3300, 3214, 3105 (N–H), 1677 (C=O), 1651 (C=N), 1603, 1563, 1486 (Ar), 1530, 1346 (-NO_2_), 1391, 1189 (O=S=O), 627 (C–S–C); ^1^H-NMR (600 MHz, CDCl_3_) δ: 8.39–8.35 (m, 2H, C_20_–H, C_22_–H), 8.14–8.10 (m, 2H, C_19_–H, C_23_–H), 7.92 (d, *J* = 8.1 Hz, 1H, C_8_–H), 7.26 (d, *J* = 1.6 Hz, 1H, C_5_–H), 7.22 (dd, *J* = 8.1, 1.5 Hz, 1H, C_7_–H), 6.22 (s, 2H, -NH_2_), 4.79 (dd, *J* = 13.9, 4.9 Hz, 1H, C_2_–H), 2.98 (m, 1H, C_11_–H), 2.41–2.25 (m, 2H, C_3_–H), 1.48 (s, 3H, C_4_–CH_3_), 1.43 (s, 3H, C_4_–CH_3_), 1.29 (d, *J* = 6.9 Hz, 6H, C_11_–CH_3_); ^13^C-NMR (151 MHz, CDCl_3_) δ: 192.84, 162.88, 156.63, 156.14, 151.50, 151.25, 141.87, 129.57, 128.50, 128.19, 125.10, 124.64, 124.10, 49.85, 44.70, 35.74, 34.66, 30.60, 29.47, 23.66; MS (ESI) *m*/*z* = 516.08 ([M + H^+^]).

*2-((5-Amino-1-((4-methyphenyl)sulfonyl)-1H-1,2,4-triazol-3-yl)thio)-6-isopropyl-4,4-dimethyl-3,4-dihydronaphthalen-1(2H)-one* (**6m**): White solid, m.p. 196.9–198.3 °C; FT-IR (KBr, cm^−1^): 3457, 3300, 3218, 3100 (N–H), 1682 (C=O), 1649 (C=N), 1603, 1561, 1486 (Ar), 1367, 1183 (O=S=O), 604 (C–S–C); ^1^H-NMR (600 MHz, DMSO*-d_6_*) δ: 7.79 (t, *J* = 8.1 Hz, 3H, C_8_–H, C_19_–H, C_23_–H), 7.52–7.43 (m, 4H, C_20_–H, C_22_–H, -NH_2_), 7.41 (s, 1H, C_5_–H), 7.27 (d, *J* = 8.1 Hz, 1H, C_7_–H), 4.91 (dd, *J* = 13.8, 4.7 Hz, 1H, C_2_–H), 3.01–2.94 (m, 1H, C_11_–H), 2.41 (s, 3H, C_21_–CH_3_), 2.19 (m, 2H, C_3_–H), 1.37 (d, *J* = 23.2 Hz, 6H, C_4_–CH_3_), 1.22 (d, *J* = 6.9 Hz, 6H, C_11_–CH_3_); ^13^C-NMR (151 MHz, DMSO-*d_6_*) δ: 193.21, 161.63, 158.02, 155.90, 152.05, 146.75, 133.71, 130.80, 128.69, 127.97, 127.73, 125.12, 124.84, 49.93, 44.74, 35.86, 34.31, 30.56, 29.28, 23.94, 23.93, 21.66; MS (ESI) *m*/*z* = 485.17 ([M + H^+^]).

*2-((5-Amino-1-((2-chlorophenyl)sulfonyl)-1H-1,2,4-triazol-3-yl)thio)-6-isopropyl-4,4-dimethyl-3,4-dihydronaphthalen-1(2H)-one* (**6n**): White solid, m.p. 195.1–194.5 °C; FT-IR (KBr, cm^−1^): 3445, 3304, 3222, 3148 (N–H), 1682 (C=O), 1649 (C=N), 1603, 1565, 1485 (Ar), 1368, 1188 (O=S=O), 614 (C–S–C); ^1^H-NMR (600 MHz, CDCl_3_) δ: 8.24 (dd, *J* = 8.0, 1.4 Hz, 1H, C_23_–H), 7.91 (d, *J* = 8.1 Hz, 1H, C_8_–H), 7.63 (m, 1H, C_21_–H), 7.57 (dd, *J* = 7.9, 0.9 Hz, 1H, C_20_–H), 7.51-7.45 (m, 1H, C_22_–H), 7.20 (s,1H, C_5_–H), 7.18 (dd, *J* = 8.1,1.4Hz,1H, C_7_–H), 6.31 (d, *J* = 25.6 Hz, 2H, -NH_2_), 4.81 (dd, *J* = 14.0, 4.6 Hz, 1H, C_2_–H), 3.00-2.90 (m, 1H, C_11_–H), 2.30-2.18 (m, 2H, C_3_–H), 1.33 (s, 3H, C_4_–CH_3_), 1.28 (s, 3H, C_4_–CH_3_), 1.27 (d, *J* = 4.5 Hz, 6H, C_11_–CH_3_); ^13^C-NMR (151 MHz, CDCl_3_) δ: 193.34, 161.55, 157.27, 155.84, 151.59, 135.77, 134.42, 133.25, 132.81, 132.21, 128.50, 128.06, 127.34, 124.87, 124.01, 50.47, 45.04, 35.67, 34.61, 30.56, 29.03, 23.66, 23.63; MS (ESI) *m*/*z* = 505.12 ([M + H^+^]).

*2-((5-Amino-1-((4-bromophenyl)sulfonyl)-1H-1,2,4-triazol-3-yl)thio)-6-isopropyl-4,4-dimethyl-3,4-dihydronaphthalen-1(2H)-one* (**6o**): White solid, m.p. 203.8–204.9 °C; FT-IR (KBr, cm^−1^): 3467, 3306, 3212, 3095 (N–H), 1676 (C=O), 1652 (C=N), 1603, 1573, 1487 (Ar), 1382, 1186 (O=S=O), 619 (C–S–C); ^1^H-NMR (600 MHz, DMSO*-d_6_*) δ: 7.91 (dd, *J* = 5.2, 3.4 Hz, 2H, C_19_–H, C_23_–H), 7.82 (d, *J* = 7.5 Hz, 2H, C_20_–H, C_22_–H), 7.78 (d, *J* = 8.1 Hz, 1H, C_8_–H), 7.55 (s, 2H, -NH_2_), 7.40 (s, 1H, C_5_–H), 7.31-7.24 (m, 1H, C_7_–H), 4.91 (dd, *J* = 13.3, 5.3 Hz, 1H, C_2_–H), 3.03-2.93 (m, 1H, C_11_–H), 2.19 (t, *J* = 9.5 Hz, 2H, C_3_–H), 1.36 (d, *J* = 14.4 Hz, 6H, C_4_–CH_3_), 1.22 (d, *J* = 6.8 Hz, 6H, C_11_–CH_3_); ^13^C-NMR (151 MHz, DMSO*-d_6_*) δ: 193.14, 162.18, 158.11, 155.91, 152.03, 135.72, 133.55, 130.12, 129.82, 128.67, 127.73, 125.14, 124.86, 49.93, 44.73, 35.86, 34.31, 30.59, 29.29, 23.94; MS (ESI) *m*/*z* = 548.07 ([M + H^+^]).

*2-((5-Amino-1-((2,4-difluorophenyl)sulfonyl)-1H-1,2,4-triazol-3-yl)thio)-6-isopropyl-4,4-dimethyl-3,4-dihydronaphthalen-1(2H)-one* (**6p**): White solid, m.p. 177.0–177.7 °C; FT-IR (KBr, cm^−1^): 3434, 3306, 3224, 3157 (N–H), 1682 (C=O), 1650 (C=N), 1603, 1565, 1488 (Ar), 1385, 1191 (O=S=O), 622 (C–S–C); ^1^H-NMR (600 MHz, CDCl_3_) δ: 8.02 (m, 1H, C_23_–H), 7.90 (d, *J* = 8.1 Hz, 1H, C_8_–H), 7.22 (d, *J* = 1.4 Hz, 1H, C_5_–H), 7.18 (dd, *J* = 8.1, 1.5 Hz, 1H, C_7_–H), 7.08–7.02 (m, 1H, C_22_–H), 6.98 (m, 1H, C_20_–H), 6.25 (s, 2H, -NH_2_), 4.83 (dd, *J* = 14.0, 4.6 Hz, 1H, C_2_–H), 2.95 (m, 1H, C_11_–H), 2.40-2.19 (m, 2H, C_3_–H), 1.38 (d, *J* = 1.2 Hz, 6H, C_4_–CH_3_), 1.27 (d, *J* = 6.9 Hz, 6H, C_11_–CH_3_); ^13^C-NMR (151 MHz, CDCl_3_) δ: 193.18, 168.21, 168.13, 166.47, 166.40, 162.20, 161.38, 161.29, 159.64, 159.5, 156.80, 155.91, 151.57, 133.46, 133.39, 128.51, 128.10, 124.91, 124.05, 121.34, 121.31, 121.25, 121.23, 112.64, 112.62, 112.49, 112.47, 106.46, 106.30, 106.29, 106.13, 50.27, 45.01, 35.71, 34.63, 30.59, 29.17, 23.67, 23.64; MS (ESI) *m*/*z* = 507.07 ([M + H^+^]).

*2-((5-Amino-1-((4-nitrilephenyl)sulfonyl)-1H-1,2,4-triazol-3-yl)thio)-6-isopropyl-4,4-dimethyl-3,4-dihydronaphthalen-1(2H)-one* (**6q**): White solid, m.p. 184.8–185.5 °C; FT-IR (KBr, cm^−1^): 3464, 3306, 3226, 3106 (N–H), 2236 (w, –CN), 1674 (C=O), 1654 (C=N), 1602, 1567, 1493 (Ar), 1389, 1190 (O=S=O), 644 (C–S–C); ^1^H-NMR (600 MHz, CDCl_3_) δ: 8.04 (d, *J* = 8.6 Hz, 2H, C_19_–H, C_23_–H), 7.93 (d, *J* = 8.1 Hz, 1H, C_8_–H), 7.83 (d, *J* = 8.6 Hz, 2H, C_20_–H, C_22_–H), 7.26 (d, *J* = 1.3 Hz, 1H, C_5_–H), 7.22 (dd, *J* = 8.1, 1.4 Hz, 1H, C_7_–H), 6.31 (s, 2H, -NH_2_), 4.80 (dd, *J* = 13.9, 4.8 Hz, 1H, C_2_–H), 3.02-2.93 (m, 1H, C_11_–H), 2.39-2.25 (m, 2H, C_3_–H), 1.47 (s, 3H, C_4_–CH_3_), 1.44 (s, 3H, C_4_–CH_3_), 1.29 (d, *J* = 6.9 Hz, 6H, C_11_–CH_3_); ^13^C-NMR (151 MHz, CDCl_3_) δ: 192.90, 162.67, 156.71, 156.14, 151.52, 140.47, 133.16, 128.75, 128.49, 128.19, 125.09, 124.11, 118.57, 116.76, 49.91, 44.74, 35.74, 34.66, 30.64, 29.45, 23.68, 23.67; MS (ESI) *m*/*z* = 496.17 ([M + H^+^]).

### 3.8. In Vitro Antiproliferative Evaluation

All the synthesized compounds were evaluated for their in vitro cytotoxicity against the human cancer cell lines T-24, MCF-7, HepG2, A549, and HT-29, by MTT assay. The cell lines were plated in 96-well plates at a density of 5 × 10^3^ cells/well and maintained at 37 °C with 5% CO_2_. The synthesized compounds were dissolved in DMSO, and further dilutions were made with DMEM, with 5-FU as the positive control. The concentrations of the compounds used were 0.8, 4, 20, 100 µM. After treatment for another 48 h and 72 h with different concentrations of the samples (0.8, 4, 20, 100 µM), 20 µL of MTT (5 mg/mL) was added and incubated for about 4 h. The medium was thrown away and the purple formazan precipitations were dissolved in 150 µL DMSO. The O. D. value was measured at 490 nm using an enzyme labeling instrument [24].

## 4. Conclusions

Seventeen novel longifolene-derived tetralone derivatives bearing 1,2,4-triazole heterocycle were designed and synthesized from longifolene. Their structures were confirmed by multiple techniques. The synthesized compounds were screened for cytotoxic activity against a panel of five human cancer cell lines using MTT assay. Some compounds exhibited better anticancer activity against the tested cancer cell lines compared to positive control 5-FU. Some intriguing structure–activity relationships were found and discussed by theoretical calculation. Compounds **6g**, **6h**, and **6d**, with excellent and broad-spectrum antitumor activity against almost all the tested cancer cell lines, were leading compounds for further investigation.

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
