# Peer review of "Synthesis and Antiproliferative Evaluation of Novel Longifolene-Derived Tetralone Derivatives Bearing 1,2,4-Triazole Moiety"

_molecules, 2020, doi:10.3390/molecules25040986_

Round 1

Reviewer 1 Report

The manuscript describes the preparation of several simple to obtain tetralone-triazole-thioethers derivatives, with potential anti-proliferative activity against several human cell lines, some some comparable to 5-fluorouracil. Computational calculations were used to justify the activity of one of the most active compounds, in comparison with a couple of other, and offer an explanation based on calculated electrostatic potentials and polarities. 

Please take note of the following issues, that must be addressed. I have some concerns on the structural assignment you made using the available data. Not that I disagree with your conclusions, but rather that a more solid evidence should be made available. Please read below. 

Lines 43-57:

No reference is given in the main text for the preparation of 2, 3 and 4, which are your starting materials, so this is crucial. A reference 31 is cited later on (vide infra). 

Lines 58, 60, 69, 72, 160:

Compounds 2, 3, 4 and are not intermediates. Please correct all occurrences. 

Lines 71-72:

Which target compounds did you compared compound 5 1H NMR with? If those compounds are known and their spectra published, you did not listed the references. 

Line 74:

The NOESY spectrum provided in the Supple. Material for compound 6a shows many artifacts, and T1 noise which obscure and complicate its interpretation, and makes unreliable assignments made using it. The selection of a small area of that spectrum, which you used as a proof of the correct assignment of the compound is consequently very risky! Perhaps a 1D NOE difference spectrum could be a safer alternative, or even an HMBC 2D spectrum? BTW, it seems like Figure S4 in the supplementary material shows an HMBC instead of an HMQC, as stated in that section. Please correct that, too! 

Lines 145 and following (Experimental section):

It is expected you to detail the experimental procedures used, no just list the spectral data obtained. 

Line 159:

If you followed a previously published procedure, please indicate the amounts used and obtained, and how exactly you prepared or obtained the solvents and reagents (in this case, the catalyst preparation IS NOT explicated in the reference cited, but in a reference cited on it, instead) A reader should be able to repeat your procedure following the experimental details. 

Author Response

Dear Reviewer,

        The manuscript (molecules-725418) has been revised using the "Track Changes" function in Microsoft Word according to the comments of the reviewers. We also responded point by point to the reviewer’s comments as listed below.

        Hope these will make it more acceptable for publication.

        Thank you.

Sincerely yours,

Wengui Duan

Feb. 20, 2020

Response to Reviewer #1

Please take note of the following issues, that must be addressed. I have some concerns on the structural assignment you made using the available data. Not that I disagree with your conclusions, but rather that a more solid evidence should be made available. Please read below.

Question 1: Lines 43-57: No reference is given in the main text for the preparation of 2, 3 and 4, which are your starting materials, so this is crucial. A reference 31 is cited later on (vide infra).

Answer: According to the comment of reviewer, the corresponding references of compounds (2, 3 and 4) were added to the section of synthesis and references in the manuscript, respectively.

  1. Tyagi, B.; Mishra, M.K.; Jasra, R.V. Solvent free synthesis of 7–isopropyl–1,1–dimethyltetralin by the rearrangement of longifolene using nano–crystalline sulfated zirconia catalyst. J. Mol. Catal. A–Chem. 2009, 301, 67–78.

  1. Liu, Z.; Cheng, Z.; Wang, C.G. Process for preparing a floral odorous perfume and perfume obtained thereform. U.S. Patent No. 5,817,616. 6 Oct. 1998.

  1. Tanemura, K.; Suzuki, T.; Nishida, Y.; Satsumabayashi, K.; Horaguchi, T. A mild and efficient procedure for α-bromination of ketones using N-bromosuccinimide catalysed by ammonium acetate. Chem. Commun. 2004, 4, 470-471.

Question 2: Lines 58, 60, 69, 72, 160: Compounds 2, 3, 4 and 5 are not intermediates. Please correct all occurrences.

Answer: The related contents have been revised.

Question 3: Lines 71-72: Which target compounds did you compared compound 5 1H NMR with? If those compounds are known and their spectra published, you did not listed the references.

Answer: Thanks for your reminder. We compared the 1H-NMR spectra of compound 5 with all the target compounds. The compounds 4, 5, and 6 are new.

Question 4: Line 74: The NOESY spectrum provided in the Supple. Material for compound 6a shows many artifacts, and T1 noise which obscure and complicate its interpretation, and makes unreliable assignments made using it. The selection of a small area of that spectrum, which you used as a proof of the correct assignment of the compound is consequently very risky! Perhaps a 1D NOE difference spectrum could be a safer alternative, or even an HMBC 2D spectrum? BTW, it seems like Figure S4 in the supplementary material shows an HMBC instead of an HMQC, as stated in that section. Please correct that, too!

Answer: Thank you for your advice. The NOESY spectrum of compound 6a was the original spectrum without attenuating in the supplementary material. When the signal in the original NOESY spectrum was simultaneously decayed to an appropriate level, the correlation signal between the amino group (4H-site) and methyl groups (C19-CH3, C23-CH3) on the benzene ring was remained. In other words, the hydrogen atoms of the amino group showed a stronger correlation with the hydrogen atoms of the methyl group on the phenylsulfonyl moiety than that of the methyl groups on the tetralone moiety, indicating the amino group was closer to the phenylsulfonyl group than the methyl groups on the tetralone moiety. In the manuscript, the NOESY spectrum signal was attenuated to an appropriate level. The related contents of the supplementary material have been revised.

Question 5: Lines 145 and following (Experimental section): It is expected you to detail the experimental procedures used, no just list the spectral data obtained.

Answer: The procedure for the synthesis of the compounds 3, 4, 5, and 6a-6q were listed at experimental section 3.4, 3.5, 3.6, and 3.7, respectively.

Question 6: Line 159: If you followed a previously published procedure, please indicate the amounts used and obtained, and how exactly you prepared or obtained the solvents and reagents (in this case, the catalyst preparation IS NOT explicated in the reference cited, but in a reference cited on it, instead) A reader should be able to repeat your procedure following the experimental details.

Answer: The catalyst was synthesized according to previous reports. The related contents and corresponding references have been added to the manuscript.

[31] Tyagi, B.; Mishra, M.K.; Jasra, R.V. Synthesis of 7-substituted 4-methyl coumarins by pechmann reaction using nano-crystalline sulfated-zirconia. Mol. Catal. A: Chem. 2007, 276, 47-56.

Reviewer 2 Report

The research article entitled “Synthesis and Antiproliferative Evaluation of Novel Longifolene-Derived Tetralone Derivatives Bearing 1,2,4-Triazole Moiety” deals with synthesis and characterization of new compounds with a tetrahydronaphatlene structure derived from the abundant and naturally longifolene. The in vitro cytotoxicity of these compounds are evaluated against a panel of five human cancer cell line using 5-fluorouracil as positive control. The manuscript is very interesting and well done but it should be revised in any aspects. The English language is corrected.

The reviewer thinks that this research article is suitable for publication in Molecules after major revision (below reported):

  • In the synthesis section (pag.4, line 54-60), the authors must report the corresponding reference of the known compounds (2 and 3) and in the references section (pag.11-12). Additionally, the characterization of these known compounds (pag. 5-6) is not necessary, so please delete their characterization from the main manuscript or move them in the supplementary materials.
  • In the Scheme 1, please insert a table with the number of compounds and their corresponding R. Using a table in the scheme is easier to understand than a list.
  • Line 69-71, pag.2: the tautomeric forms of intermediate 5 is an important chemical aspect, so please explain better this concept and highlight the importance and the influence of the reaction conditions in the tautomerism. About this, did the authors observe only one compound (1H-form) or also the other isomers in low yield?? If yes, please insert this information in the manuscript.
  • The authors should make some general corrections (in the manuscript and on the supplemental materials): change “1H NMR” with “1H-NMR”; “13C NMR” with “13C-NMR”; “IR” with “FT-IR”; “C = O, C = N” with “C=O, C=N”; “DMDSO-d6” with “DMSO-d6” (in supplementary material).
  • Line 97, pag.3: the authors affirm that the in vitro anti-proliferative activity of intermediates 2, 3, 4 and 5 was evaluated, but these data is not showed in the table 1. Please insert the biological activity of the intermediates 2, 3 ,4 and 5 in the table 1.
  • Table 1, compound 6d: please change “R= R= 3’-Br” with “R = 3’-Br”
  • Table 1, compound 6g: the activity against MCF-7 cell is not correct. The SD is too higher (4.42±93), please test again compound 6g and check the data.
  • Line 109, pag.4: compound 6e shows an IC50 > 100 µM against HT-29 and IC50= 82.36 µM against A549. Based on the data showed, compound 6e doesn’t show a good antitumor activity as the other compounds mentioned (6b, 6c and 6f). So please, delete compound 6e in this phrase.
  • Line 114-120, pag.4: the structure-activity relationships (SARs) are like a list of considerations, not real SARs. Please improve this part and try to give conclusions more detailed.
  • Line 136, pag. 5: please change “(Fig. 2, b, e, h)” with “(Fig. 2, b, e)”.
  • Line 144, pag. 5: please change “6m” with “6j”.
  • Line 185, pag. 6: the authors write “the filtrate was washed with water, dried and evaporated”, so if the compound is solid what do you evaporate? Please check this affirmation.
  • Line 187, pag. 6: the melting point of compound 4 is missing. Please insert it.  

Author Response

Dear Reviewer,

        The manuscript (molecules-725418) has been revised using the "Track Changes" function in Microsoft Word according to the comments of the reviewers. We also responded point by point to the reviewer’s comments as listed below.

        Hope these will make it more acceptable for publication.

        Thank you.

Sincerely yours,

Wengui Duan

Feb. 20, 2020

Response to Reviewer #2

The reviewer thinks that this research article is suitable for publication in Molecules after major revision (below reported):

Question 1: In the synthesis section (pag.4, line 54-60), the authors must report the corresponding reference of the known compounds (2 and 3) and in the references section (pag.11-12). Additionally, the characterization of these known compounds (pag. 5-6) is not necessary, so please delete their characterization from the main manuscript or move them in the supplementary materials.

Answer: According to the comment of reviewer, the corresponding references of compounds (2 and 3) were added to the synthesis section and references in the manuscript. The characterization data of compounds (2 and 3) were deleted from the main text of the manuscript.

Question 2: In the Scheme 1, please insert a table with the number of compounds and their corresponding R. Using a table in the scheme is easier to understand than a list.

Answer: The table for the R groups of the target compounds has been inserted in the Scheme 1.

Question 3: Line 69-71, pag.2: the tautomeric forms of intermediate 5 is an important chemical aspect, so please explain better this concept and highlight the importance and the influence of the reaction conditions in the tautomerism. About this, did the authors observe only one compound (1H-form) or also the other isomers in low yield?? If yes, please insert this information in the manuscript.

Answer: We tracked the N-sulfonylation reaction process of compound 5 by HPLC, and only one new peak was observed. Therefore, the compound 5 was speculated to be 1H-forms referring to the report. This result was confirmed by comparing the 1H-NMR spectra of compound 5 with all the target compounds and the NOESY spectrum of compound 6a. The related information has been inserted in the manuscript.

Question 4: The authors should make some general corrections (in the manuscript and on the supplemental materials): change “1H NMR” with “1H-NMR”; “13C NMR” with “13C-NMR”; “IR” with “FT-IR”; “C = O, C = N” with “C=O, C=N”; “DMDSO-d6” with “DMSO-d6” (in supplementary material).

Answer: The related contents have been revised.

Question 5: Line 97, pag.3: the authors affirm that the in vitro anti-proliferative activity of intermediates 2, 3, 4 and 5 was evaluated, but these data is not showed in the table 1. Please insert the biological activity of the intermediates 2, 3 ,4 and 5 in the table 1.

Answer: Thanks for your reminder. The in vitro anti-proliferative activities of intermediates 2, 3, 4 and 5 were showed in Table 2. The related contents have been revised in manuscript.

Question 6: Table 1, compound 6d: please change “R= R= 3’-Br” with “R = 3’- Br”

Answer: The related content has been revised.

Question 7: Table 1, compound 6g: the activity against MCF-7 cell is not correct. The SD is too higher (4.42±93), please test again compound 6g and check the data.

Answer: After repeated confirmation, the IC50 value of compound 6g for MCF-7 was 4.42 ± 2.93 µM. The IC50 value has been revised.

Question 8: Line 109, pag.4: compound 6e shows an IC50 > 100 µM against HT-29 and IC50= 82.36 µM against A549. Based on the data showed, compound 6e doesn’t show a good antitumor activity as the other compounds mentioned (6b, 6c and 6f). So please, delete compound 6e in this phrase.

Answer: The related content has been revised.

Question 9: Line 114-120, pag.4: the structure-activity relationships (SARs) are like a list of considerations, not real SARs. Please improve this part and try to give conclusions more detailed.

Answer: Thank you for this enlightening comment. The expression of the “the structure-activity relationships” has been changed as “Some intriguing bioactive differences”.

Question 10: Line 136, pag. 5: please change “(Fig. 2, b, e, h)” with “(Fig. 2, b, e)”.

Answer: The related content has been revised.

Question 11: Line 144, pag. 5: please change “6m” with “6j”.

Answer: The “6m” has been revised to “6j”.

Question 12: Line 185, pag. 6: the authors write “the filtrate was washed with water, dried and evaporated”, so if the compound is solid what do you evaporate? Please check this affirmation.

Answer: Thanks for your reminder. The related content has been revised to "The filtrate was washed with water, dried, and evaporated to remove Et2O."

Question 13: Line 187, pag. 6: the melting point of compound 4 is missing. Please insert it.

Answer: The m.p. value has been added.

Round 2

Reviewer 2 Report

Thank you for your corrections. I have seen a mistake in the scheme 1: Please correct "6a-6g" with "6a-6q" !!!!!!!